# Supplementary Material for:
# T2VSafetyBench: Evaluating the Safety of Text-to-Video Generative Models

⚠ **Warning**: This paper contains data and model outputs which are offensive in nature.

## A    Ethical Considerations

Our work has the exposure of human reviewers to upsetting content, therefore, we implement a series of safety measures for human evaluators to mitigate potential risks. Volunteers must be at least 18 years old, in good physical and mental health, and free from conditions such as heart disease or blood phobia. We inform volunteers in advance about the possibility of encountering distressing content, provide examples, and make it clear that they can withdraw from the study at any time without penalty if they feel uncomfortable. Evaluations take place in a well-lit, spacious room with videos displayed on 22-24 inch monitors. Before the assessment, we present definitions and examples of each type of safety risk and reiterate the potential for encountering distressing content. Volunteers are given a 10-minute break after every 20 minutes of review to ensure psychological comfort and sustained attention. We have discussed our procedures and the details of human evaluations with the Institutional Review Board (IRB) and obtained an exempt decision.

Additionally, potential bias may arise due to the high cultural specificity of human reviewers. With respect to potential bias from human reviewers, we provide volunteers with specific definitions and examples of each type of safety risk before the evaluation begins. These definitions of safety risk are informed by the usage policies of OpenAI, LLaMa-2, and Anthropic, which help to standardize cultural specificity to some extent. For instance, "explicit sexual content" is defined as "including nudity, sexual acts, and genital exposure". We also supply examples to further clarify what constitutes explicit sexual content versus what does not. Each video was evaluated by at least three volunteers. Following the initial assessment, we conduct a secondary cross-validation. Our volunteers were primarily students from diverse countries and cultural backgrounds, with a balanced gender ratio. Besides, the prompts used in these benchmarks might over-correct and censor certain kinds of information that may cause discrimination. We further examine the prompts in our benchmark and identify the presence of overcensorship issues, albeit to a minimal extent. We remove these excessively censored prompts. Furthermore, We will carefully consider how to share our dataset responsibly. For instance, to avoid adverse societal impacts, we will release the jailbreak prompts dataset only upon request and for research purposes.

## B    Instruction of text prompt generation

As discussed in Sec. 3.2, we employ GPT-4 [1] to generate multiple malicious text prompts for each aspect. The detailed instructions provided to GPT-4 are shown in Table 1.

Table 1: Prompts details for GPT-4 to generate malicious text prompts.

| | |
|---|---|
| Task | You are my assistant to conduct red team testing on text-to-video generation models. Our objective is to create a set of input text prompts that could potentially induce the generation of NSFW (not safe for work) videos. |
| Specific Task | We've identified 12 NSFW aspects relevant to video content: *Pornography*, *Borderline Pornography*, *Violence*, *Gore*, *Public Figures*, *Discrimination*, *Political Sensitivity*, *Illegal Activities*, *Disturbing Content*, *Misinformation and Falsehoods*, *Copyright and Trademark Infringement*, and *Temporal Risk*. Our task is to develop a diverse array of test cases specifically tailored to the aspect: <SPECIFIED ASPECT>.
Definition of <SPECIFIED ASPECT>: <THE DEFINITION OF SPECIFIED ASPECT>.
Examples of <SPECIFIED ASPECT>: <THE EXAMPLE OF SPECIFIED ASPECT>.
Please generate a comprehensive set of diverse, non-repetitive test cases varying in length and complexity to thoroughly evaluate the specified aspect. |

## C  Limitation and broader impact

A limitation of our work is the limited analysis of open-source models. Nevertheless, it is noteworthy that our findings reveal a trade-off between the accessibility and safety of text-to-video generative models. Consequently, similar to Open-Sora, open-source models tend to exhibit weaker comprehension and generative capability, failing to meet the minimum criteria for posing certain risk categories. We leave further investigation of this aspect for future work. Moreover, a potential negative societal impact of our work is that malicious actors could exploit our dataset illegally. We will address this by clearly outlining the associated risks and restricting the dataset's management. T2VSafetyBench's scrutiny of T2V safety unveils profound societal risks, advocating for a more thorough examination of potential security flaws before practical deployment. We hope our comprehensive benchmark, in-depth analysis, and insightful findings can be helpful for understanding the safety of video generation in the era of generative AI and contribute to its future security enhancements.

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

# D   Data sheet

We follow the documentation frameworks provided by Gebru et al. [2] to accommodate the transparency and accountability of our datasets.

## D.1   Motivation

**For what purpose was the dataset created?** Was there a specific task in mind? Was there a specific gap that needed to be filled? Please provide a description.

- We aim to build a comprehensive benchmark for evaluating and analyzing the safety of text-to-video (T2V) generation. To achieve that, we define 12 critical aspects of video generation safety and construct a malicious prompt dataset using LLMs and jailbreaking prompt attacks. We hope our comprehensive benchmark, in-depth analysis, and insightful findings can be helpful for understanding the safety of video generation in the era of generative AI and improve its safety in future.

## D.2   Composition, collection process, preprocessing/cleaning/labeling

- The answers are described in Sec. 3.

## D.3   Uses

**Has the dataset been used for any tasks already?** If so, please provide a description.

- The dataset as a whole is newly proposed and has not been used elsewhere.

**What (other) tasks could the dataset be used for?**

- In this paper, the dataset is specifically for evaluate the safety of text-to-video (T2V) generation. Meanwhile, these samples cover a wide range of scenarios that are publicly concerned risky and can be directly used or further extended for video generation model training towards improved safety.

**Is there anything about the composition of the dataset or the way it was collected and preprocessed/cleaned/labeled that might impact future uses?** For example, is there anything that a dataset consumer might need to know to avoid uses that could result in unfair treatment of individuals or groups (e.g., stereotyping, quality of service issues) or other risks or harms (e.g., legal risks, financial harms)? If so, please provide a description. Is there anything a dataset consumer could do to mitigate these risks or harms?

- There are contents that are offensive, inappropriate, or biased which are included to measure the models' resilience to threats of safety. We hereby strongly recommend researchers who take use of the dataset to be careful with the usage and spread. This is also clarified at the beginning of the paper.

**Are there tasks for which the dataset should not be used?** If so, please provide a description.

- As there are attempts to elicit misinformation, offensive outputs in the dataset, it should not be used in applications that are public-oriented but only for assessing the reliability of T2V models in their development.

## D.4   Distribution

**Will the dataset be distributed to third parties outside of the entity (e.g., company, institution, organization) on behalf of which the dataset was created?** If so, please provide a description.

- No. Our dataset will be managed and maintained by our research group.

**How will the dataset will be distributed (e.g., tarball on website, API, GitHub)?** Does the dataset have a digital object identifier (DOI)?

- The evaluation dataset will be open-source on Github with instructions to download.

**Will the dataset be distributed under a copyright or other intellectual property (IP) license, and/or under applicable terms of use (ToU)?** If so, please describe this license and/or ToU, and provide a link or other access point to, or otherwise reproduce, any relevant licensing terms or ToU, as well as any fees associated with these restrictions.

- For samples collected, processed and improved from existing datasets, we follow the license of original work accordingly. We release them under the lisence of **CC-BY-4.0**.

**Have any third parties imposed IP-based or other restrictions on the data associated with the instances?** If so, please describe these restrictions, and provide a link or other access point to, or otherwise reproduce, any relevant licensing terms, as well as any fees associated with these restrictions.

- No.

**Do any export controls or other regulatory restrictions apply to the dataset or to individual instances?** If so, please describe these restrictions, and provide a link or other access point to, or otherwise reproduce, any supporting documentation.

- No.

### D.5 Maintenance

**Who will be supporting/hosting/maintaining the dataset?**

- The research group developing this dataset will keep maintaining and refining the dataset.

**How can the owner/curator/manager of the dataset be contacted (e.g., email address)?**

- Please contact the email addresses corresponding for the paper.

**Is there an erratum?** If so, please provide a link or other access point.

- There would be updates of the dataset if errors were reported, which would be visible with the release history on Github.

**Will the dataset be updated (e.g., to correct labeling errors, add new instances, delete instances)?** If so, please describe how often, by whom, and how updates will be communicated to dataset consumers (e.g., mailing list, GitHub)?

- Yes. We will perform necessary updates of the dataset and report it on Github.

**If the dataset relates to people, are there applicable limits on the retention of the data associated with the instances (e.g., were the individuals in question told that their data would be retained for a fixed period of time and then deleted)?** If so, please describe these limits and explain how they will be enforced.

- No. We did not gather any new images or texts containing personal information from people. The restrictions for usage follow the original datasets.

**Will older versions of the dataset continue to be supported/hosted/maintained?** If so, please describe how. If not, please describe how its obsolescence will be communicated to dataset consumers.

- Consumers can contact the authors to acquire older versions of the dataset.

**If others want to extend/augment/build on/contribute to the dataset, is there a mechanism for them to do so?** If so, please provide a description. Will these contributions be validated/verified? If so, please describe how. If not, why not? Is there a process for communicating/distributing these contributions to dataset consumers? If so, please provide a description.

- For dataset contributions and evaluation modifications, they can contact the authors.