# OpenReview forum: "T2VSafetyBench: Evaluating the Safety of Text-to-Video Generative Models"
_NeurIPS.cc/2024/Datasets_and_Benchmarks_Track — NeurIPS 2024 Track Datasets and Benchmarks Poster_

### Official Review · Reviewer_zpBN · 2024-07-10
**Good paper but may be improved**

**Rating:** 7
**Confidence:** 5
**Correctness:** Yes
**Clarity:** Yes

**Review:**

Pros:
The authors focus on the new perspectives of evualtion of t2v models.
The paper is easy to understand and well-written.


Cons:
First, According to the official requriment: *For benchmarks, the supplementary materials must ensure that all results are easily reproducible. Where possible, use a reproducibility framework such as the ML reproducibility checklist, or otherwise guarantee that all results can be easily reproduced, i.e. all necessary datasets, code, and evaluation procedures must be accessible and documented.*
I do not find any code for your benchmark. Also, I cannot reproduce your benchmark with current information. Therefore, I consider this as an incomplete submission.

Second, the number of current prompts is too small. As shown in the Supplementary, I only find less than 100 prompts for each class. What’s more, they are all generated by GPT. They may fail to reflect the prompts in the real-world.

It seems the paper does not show insight from evaluating existing t2v models. Without checking the experimental results, I can expect the results. I prefer to see more interesting findings.

**Strengths:**

The authors focus on the new perspectives of evualtion of t2v models.
The paper is easy to understand and well-written.

**Additional Feedback:**

N/A

**Documentation:**

No

**Ethics:**

No.

**Limitations:**

Yes

**Opportunities For Improvement:**

See Cons.

**Relation To Prior Work:**

Yes

**Summary And Contributions:**

The paper proposes a benchmark for evaluating the safety of text-to-video models. Specifically, the authors list 12 different evaluation aspects for evaluation.

---

> ### Author Rebuttal · Authors · 2024-08-26
>
> **We apologize for the delay in submitting our rebuttal. The delay is due to our efforts to incorporate additional experiments based on the reviewers' feedback, which have only recently been completed.**
> Thank you for appreciating our new contributions as well as providing the valuable feedback. Below we address the detailed comments, and hope that you can find our response satisfactory.
>
>
>
>
>
>
> ***Question 1: I do not find any code for your benchmark. Also, I cannot reproduce your benchmark with current information.***
>
> Thank you for your note.
> The code for T2VSafetyBench is available at the following link (https://anonymous.4open.science/r/T2VSafetyBench_Code-1763/) for reproduction purposes.
> We will provide a more detailed reproducibility checklist in the revision.
>
>
>
>
>
>
>
>
>
>
> ***Question 2: The number of current prompts is too small. What’s more, they are all generated by GPT. They may fail to reflect the prompts in the real-world.***
>
> Thanks for the valuable suggestion.
> In the Supplementary, we provide only a subset of prompts as examples.
> Additionally, we further expand the dataset to increase both the quantity and diversity of prompts by incorporating non-LLM-generated examples using VidProM [1] and Self-Instruct [2].
> VidProM is a large-scale dataset comprising unique text-to-video prompts from real users.
> We select NSFW prompts from VidProM to supplement T2VSafetyBench, thereby increasing the data sources and better reflecting the prompts in the real-world.
> Furthermore, we also employ the Self-Instruct framework with T2VSafetyBench as the seed set (where T2VSafetyBench already includes some prompts from VidProM, thus containing both real-world prompts and LLM-generated prompts), to generate a broader and more diverse range of prompts.
> The updated dataset consists of a total of 4,400 prompts, available at the following link (https://anonymous.4open.science/r/T2VSafetyBench_Dataset-248D/).
> We will add a more detailed information about dataset construction in the revision.
>
>
>
>
>
>
>
>
>
>
>
>
> ***Question 3: It seems the paper does not show insight from evaluating existing t2v models. I prefer to see more interesting findings.***
>
> We believe that **this article has two interesting findings and their underlying insights**, in addition to the conventional comparisons and analyses of safety across different dimensions.
>
> - **There is a trade-off between the usability and safety of text-to-video generative models.**
> An interesting observation is that the Pika with a non zero temporal risk profile is also the one that is claimed to be more capable of video generation.
> This phenomenon arises because Pika excels in generating continuous actions and variations unique to video content, such as complex movements, subtitle shifts, and transformations in human forms. In contrast, the other three models display weaker generative capabilities and fail to meet the minimum threshold to produce such risks.
> This observation (along with other phenomena discussed in Section 4.2) support our third conclusion: there is a trade-off between the usability and safety of text-to-video generative models.
> **This finding is novel and intriguing, as advanced generative models are generally perceived as both more performant and safer (e.g., many safety benchmarks for LLMs exhibit a high correlation with upstream model capabilities [3]). However, our results reveal this trade-off for the first time within the context of text-to-video models.**
> This implies that as models enhance, the risk of generating unsafe content may increase unless explicitly handled.
>
> - Another key finding is our second conclusion: The correlation between GPT-4’s assessments and manual reviews of text-to-video model safety is generally high. **This novel correlation is not previously identified in prior work.**
> **This discovery is significant** as it supports the rationality of leveraging GPT-4 for large-scale evaluations in our context.
>
> [1] VidProM: A Million-scale Real Prompt-Gallery Dataset for Text-to-Video Diffusion Models, arXiv 2024
>
> [2] Self-Instruct: Aligning Language Models with Self-Generated Instructions, ACL 2023
>
> [3] Safetywashing: Do AI Safety Benchmarks Actually Measure Safety Progress? arXiv 2024

---

> > ### Comment · Reviewer_zpBN · 2024-08-26
> >
> > I sincerely appreciate authors’ detailed responses, which solve most of my concerns. Therefore, I raise the score to 6 and am willing to accept this paper. An important reminder is, when publish the benchmark, please include the real-world prompts in VidProM to enhance the practicality.

---

> > > ### Author Response · Authors · 2024-08-29
> > > **Thank you for increasing the score**
> > >
> > > Dear Reviewer zpBN,
> > >
> > > Thank you very much for increasing the score and for agreeing to accept our paper! We are glad to know that our response has addressed your concerns. We really appreciate your valuable comments. We will include the real-world prompts in VidProM and improve the paper in the final version.
> > >
> > > Best regards, Authors

---

> > > > ### Comment · Reviewer_zpBN · 2024-08-29
> > > >
> > > > Thanks for this. I am still a little worried about the updated version. Please update a new version on Arxiv, and I will check that.
> > > >
> > > > Note: I come across this paper on arxiv, and this track does not require anonymous. Therefore, my requests seem do not violate the official reviewing code.

---

> > > > ### Comment · Reviewer_zpBN · 2024-08-29
> > > >
> > > > I will consider to raise score to 7 after checking the new version. There is still a 5 and I want to make sure the acceptance of this paper.

---

> > > > > ### Author Response · Authors · 2024-08-31
> > > > >
> > > > > Thank you for the follow-up comment.
> > > > > Based on all the reviewers' feedback, we have made a preliminary update to our manuscript.
> > > > > Since arXiv does not update on weekends but author/reviewer discussions end on Aug 31 '24, we list the main updates below:
> > > > >
> > > > > - **At the end of Section 1 and in Appendix A**, we add Ethical Considerations. This addresses the concerns of ethics Reviewers qvqH, dTbi, and Reviewers JP3C and sQtg regarding ethical risks.
> > > > >
> > > > > - **In Section 3.1 - Aspects**, we add a more fine-grained categorization of Temporal Risk. This update addresses the need for a more detailed discussion on Temporal Risk as noted by Reviewers JP3C and 53CX.
> > > > >
> > > > > - **In Section 3.2 - Dataset Construction**, we add Section 3.2.1 to describe the process of collecting real-world prompts from VidProM. In Section 3.2.2, we include a description of using the Self-Instruct framework.
> > > > > These updates address the concerns about dataset diversity and practicality raised by Reviewers 53CX and zpBN.
> > > > >
> > > > > - **In Section 3.3 - Evaluation Protocols**, we provide a clearer description of the evaluation process, including GPT-4 evaluation and human evaluation. This update addresses the concerns of Reviewers 4PQa and 53CX regarding the clarity and completeness of the evaluation details.
> > > > >
> > > > > - **In Section 4 - Benchmarking Results**, we include more detailed statistical information, such as video frame counts, video durations, and the number of videos generated per prompt. This update addresses the statistical information gaps pointed out by Reviewer 4PQa.
> > > > >
> > > > > - **In Section 4.3 - Discussion**, we include the discussion about safety mechanisms. This update addresses the concern about the lack of introduction to potential security mitigation strategies noted by Reviewers 4PQa and sQtg. We also add underlying insights from evaluating T2V models. This addition addresses the concerns of Reviewers sQtg and zpBN about the lack of interesting findings.
> > > > >
> > > > > - **In Appendix C**, we add experimental results on additional models. This update addresses the concern about model variety raised by reviewer 4PQa.
> > > > >
> > > > > - **In the footnote of Section 4 of the experimental section**, we add a link to the code.
> > > > >
> > > > > As arXiv does not update over the weekend, you will need to wait until next week to see the updated version.
> > > > > Given the time constraints (the author/reviewer discussions will end on Aug 31 '24, Anywhere on Earth), the current version represents a preliminary update with some areas yet to be fully revised.
> > > > > For instance, the release of dataset links is still pending.
> > > > > To prevent illegal use, we will release the prompt dataset but will provide access only upon request and for research purposes and preparations for this process are still underway.
> > > > > We will continue to improve the manuscript, code, and dataset.
> > > > > As the discussion phase is about to close, we welcome any further suggestions you might have. We are happy to continue making improvements.

---

> > > > > > ### Comment · Reviewer_zpBN · 2024-08-31
> > > > > >
> > > > > > Thanks for providing such a detailed reply. I am happy to raise the score to 7. However, it seems the editing option is closed. Therefore, I write the raising score to 7 here.

---

> > > > > > > ### Author Rebuttal · Authors · 2024-09-01
> > > > > > >
> > > > > > > The editing option may now be re-enabled or during the reviewer/AC discussion phase. Thanks.

---

> > > > > > > > ### Comment · Reviewer_zpBN · 2024-09-01
> > > > > > > >
> > > > > > > > Added to 7 :)

---

> > ### Author Response · Authors · 2024-09-01
> > **Thank you for once again increasing the score to 7!**
> >
> > Thank you very much for once again increasing the score to 7!
> >
> > Best, Authors

---

### Official Review · Reviewer_sQtg · 2024-07-20
**Review Comments**

**Rating:** 6
**Confidence:** 4
**Correctness:** Yes.
**Clarity:** Yes.

**Review:**

This paper exposes the security risks of text-to-video models and provides a comprehensive evaluation method and benchmark dataset. In particular, the examples shown illustrate the effectiveness of the method. However, there are still the following major issues that need to be addressed:

- Please note that the datasets and benchmark tracks of NeurIPS do not require anonymity, which may violate NeurIPS policy.

- The novelty of this article is limited. The methodology appears to be similar to jailbreak attacks on text-to-image models. If this is not the case, the authors should provide detailed explanations to highlight the differences.

- This article requires further approval from the ethics committee. It involves human evaluation, necessitating ethics committee approval. Additionally, the dataset contains pornographic content and human faces, which require careful handling and evaluation. The authors should exercise caution when releasing the jailbreak prompts dataset to avoid adverse societal effects.

- The dataset disclosed in this article needs further detoxification. The NSFW samples include facial information of famous people and adult content, which may confuse readers. Additionally, the jailbreak prompts need further detoxification to prevent illegal use.

- Possible security mitigation strategies need to be further provided. An introduction to potential security mitigation strategies is necessary to help the research community understand the design of security measures for text-to-video models.

**Strengths:**

+ This work offers T2VSafetyBench, a benchmark for safety-critical assessments of T2V models.
+ Extensive case studies.
+ Good writing.

**Additional Feedback:**

The ethics of this article require further review and approval.

**Documentation:**

This article discloses the jailbreak prompts dataset but does not disclose the code.

**Ethics:**

On the one hand, this article involves human evaluation, which requires the approval of the ethics committee. On the other hand, the dataset disclosed in this article involves pornographic content and human faces, which require careful handling and evaluation. The authors should be cautious about releasing the jailbreak prompts dataset to avoid adverse societal effects.

**Limitations:**

- The difference between this paper and previous work on jailbreak attacks on text-to-image models does not seem obvious and still needs further explanation.
- Note that D&B submissions do not have to be anonymized!
- The NSFW samples published need further detoxification.
- Further approval from the ethics committee is required.
- The novelty of this article is limited.

**Opportunities For Improvement:**

- Please note that the datasets and benchmark tracks of NeurIPS do not require anonymity, which may violate NeurIPS policy.

- The novelty of this article is limited. The methodology appears to be similar to jailbreak attacks on text-to-image models. If this is not the case, the authors should provide detailed explanations to highlight the differences.

- This article requires further approval from the ethics committee. It involves human evaluation, necessitating ethics committee approval. Additionally, the dataset contains pornographic content and human faces, which require careful handling and evaluation. The authors should exercise caution when releasing the jailbreak prompts dataset to avoid adverse societal effects.

- The dataset disclosed in this article needs further detoxification. The NSFW samples include facial information of famous people and adult content, which may confuse readers. Additionally, the jailbreak prompts need further detoxification to prevent illegal use.

- Possible security mitigation strategies need to be further provided. An introduction to potential security mitigation strategies is necessary to help the research community understand the design of security measures for text-to-video models.

**Relation To Prior Work:**

The difference between this paper and previous work on jailbreak attacks on text-to-image models does not seem obvious and still needs further explanation.

**Summary And Contributions:**

The development of Sora ushers in a new era in text-to-video (T2V) generation but raises significant security concerns. Generated videos might contain illegal or unethical content, and current evaluations lack comprehensive safety assessments, particularly the unique temporal risks in videos. To address this, the authors introduce T2VSafetyBench, a benchmark for safety-critical assessments of T2V models. They define 12 key safety aspects and create a malicious prompt dataset using LLMs and jailbreak prompt attacks. Key findings include: 1) no model excels in all safety aspects; 2) GPT-4 assessments correlate well with manual reviews; 3) there is a trade-off between usability and safety. As video generation advances, safety risks will increase, highlighting the urgency of prioritizing video safety. T2VSafetyBench aims to enhance the understanding of video generation safety in the AI era.

---

> ### Author Rebuttal · Authors · 2024-08-26
>
> ***Question 4: The dataset disclosed in this article needs further detoxification.***
>
> Thanks for the valuable suggestion.
> We will conduct further detoxification of the dataset before its release, following the guidelines outlined in [1], to avoid causing discomfort to readers.
> Additionally, we will carefully consider the responsible sharing of our dataset. For instance, to prevent illegal use, we will release the jailbreak prompt dataset but will provide access only upon request and for research purposes.
>
> [1] ParaDetox: Detoxification with Parallel Data, ACL 2022
>
>
>
>
>
>
>
>
>
> ***Question 5: An introduction to potential security mitigation strategies is necessary to help the research community understand the design of security measures for text-to-video models.***
>
> Thanks for the valuable suggestion.
> We provide an extended discussion about the possible security mitigation strategies here.
> Potential security mitigation strategies include detection-based methods and removal-based methods.
>
> - Detection-based methods involve filtering inappropriate content through safety classifiers. Potential options include pre-processing safety filters based on text and post-processing safety filters based on video.
>
> - Removal-based methods, on the other hand, steer the model away from undesirable content by actively guiding in inference phase or fine-tuning the model parameters.
> Potential techniques in this category include diffusion with negative prompts, concept-erased diffusion, and machine unlearning.
>
> We will add a more detailed discussion about potential security mitigation strategies in the revision.

---

> > ### Comment · Reviewer_sQtg · 2024-08-26
> > **Thanks**
> >
> > I appreciate the authors' efforts to address my concerns, and since my rating is positive, I will keep it unchanged.

---

> > > ### Author Response · Authors · 2024-08-29
> > > **Thank you for the appreciation of our contributions**
> > >
> > > Dear Reviewer sQtg,
> > >
> > > We are pleased to know that you find our response satisfactory. We really appreciate your valuable comments. We will further improve the paper in the final.
> > >
> > > Best regards, Authors

---

> ### Author Rebuttal · Authors · 2024-08-26
>
> **We apologize for the delay in submitting our rebuttal. The delay is due to our efforts to incorporate additional experiments based on the reviewers' feedback, which have only recently been completed.**
> Thank you for appreciating our new contributions as well as providing the valuable feedback. Below we address the detailed comments, and hope that you can find our response satisfactory.
>
>
>
>
>
>
>
> ***Question 1: Please note that the datasets and benchmark tracks of NeurIPS do not require anonymity, which may violate NeurIPS policy.***
>
> Thank you for your note. However, the fifth paragraph of https://nips.cc/Conferences/2024/CallForDatasetsBenchmarks contains the following bolded statement: “Authors can choose to submit either single-blind or double-blind.”
>
>
>
>
>
>
> ***Question 2: The methodology appears to be similar to jailbreak attacks on text-to-image models. If this is not the case, the authors should provide detailed explanations to highlight the differences.***
>
> Thanks for the valuable suggestion.
> We provide a detailed explanation to highlight the novelty of our work and its distinction from existing jailbreak attacks on text-to-image models.
>
> - **First, the novelty and difference of our work lie in the unique temporal risks associated with text-to-video.**
>
>   - Existing jailbreak attacks on text-to-image models have limited their focus to insufficient aspects such as pornography, violence, gore, hate, and politics, neglecting other critical safety aspects. Our work addresses these gaps by conducting a comprehensive safety assessment across 12 crucial dimensions. **Importantly, these benchmarks also fail to consider the unique temporal risks associated with video.**
>
>   - For "Temporal Risk", we propose a general definition. It encompasses any cases where "individual frames might appear harmless but the entire sequence can present unsafe content through continuity between frames". "Temporal Risk" can be classified into three types: (1) Sequential Action Risk (where individual frames extracted from actions are benign, but the continuous sequence of actions is not safe for work), (2) Dynamic Variation Risk (where unsafe content is revealed through dynamically transforming the target into another object over time), (3) Coherent Contextual Risk (where unsafe content emerges from the temporal linking of presented elements, such as harmful subtitles). The evaluation criteria for each category are consistent: “the entire sequence presents NSFW content through continuity between frames”.
>
> - **Additionally, the novelty and difference of our work lie in two important findings regarding the safety of text-to-video models, which are novel and absent in previous jailbreak attacks on text-to-image models.**
>
>   - An interesting observation is that the Pika with a non zero temporal risk profile is also the one that is claimed to be more capable of video generation. This phenomenon arises because Pika excels in generating continuous actions and variations unique to video content, such as complex movements, subtitle shifts, and transformations in human forms. In contrast, the other three models display weaker generative capabilities and fail to meet the minimum threshold to produce such risks. **This observation supports our third conclusion: there is a trade-off between the usability and safety of text-to-video generative models. This finding is novel and absent in text-to-image model jailbreak research.** This implies that as models enhance, the risk of generating unsafe content may increase unless explicitly handled.
>
>   - Another key finding is our second conclusion: The correlation between GPT-4’s assessments and manual reviews of text-to-video model safety is generally high. **This correlation is not previously identified in prior work.** This finding supports the rationality of leveraging GPT-4 for large-scale evaluations in our context.
>
> We will clarify above discussion about novelty and difference in the revision.
>
>
>
>
>
>
>
>
>
> ***Question 3: This article requires further approval from the ethics committee.***
>
> Thanks for the suggestion.
> We have provided a comprehensive Ethical Considerations in response to the comments from the two Ethics Reviewers.
>
> - Regarding the human evaluation involved in our article,
> We have implemented a series of safety measures for human evaluators to mitigate potential risks.
> Volunteers must be at least 18 years old, in good physical and mental health, and free from conditions such as heart disease or blood phobia.
> We inform volunteers in advance about the possibility of encountering distressing content, provide examples, and make it clear that they can withdraw from the study at any time without penalty if they feel uncomfortable.
> Evaluations take place in a well-lit, spacious room with videos displayed on 22-24 inch monitors.
> Prior to the assessment, we present definitions and examples of each type of safety risk and reiterate the potential for encountering distressing content.
> Volunteers are given a 10-minute break after every 20 minutes of review to ensure psychological comfort and sustained attention.
> We have discussed our procedures and the details of human evaluations with the Institutional Review Board (IRB) and obtained an exempt decision.
>
> - We will carefully consider how to share our dataset responsibly. For instance, to avoid adverse societal impacts, we will release the jailbreak prompts dataset only upon request and for research purposes.
>
> We will add the Ethical Considerations in the revision.

---

### Official Review · Reviewer_53CX · 2024-07-24
**Review comments**

**Rating:** 6
**Confidence:** 4
**Correctness:** The benchmark aligns with the claims …
**Clarity:** The paper is well written.

**Review:**

This paper focuses on a critical and timely concern in the emerging area of text-to-video generative models, which can provide a practical benchmark for developers and researchers working on the safety of T2V models.
Strengths:
1. The proposed T2VSafetyBench contains extensive aspects of safety concerns. Some aspects such as Borderline Pornography and Temporal Risk are not fully exploited yet and the introduced benchmark offers a novel research perspective for the safety of T2V models.
2. The consideration of prompt-based attacks during data construction is interesting and makes sense. Incorporating adversarial prompts can help developer better assess their model’s robustness against malicious instruction.
3. The paper is well organized with clear analysis and comprehensive discussion. Both GPT-4 and human study are provided to evaluate the four open-source models.

However, I also have the following concerns:

1.Lack of discussion and analysis on unique risks of videos. Though the benchmark focuses on the safety of T2V models, most part of the evaluation can be done by text-to-image safety benchmarks. While some unique aspects such as Temporal Risk are provided, the paper does not discuss them in detail. e.g. How many categories does Temporal Risk have? Are the evaluations for each category the same?

2.Limited sources of prompts. The prompts of T2VSafetyBench are entirely generated by GPT-4, which may limit the diversity of the assessment. Manually filtering can certainly remove abnormal prompts, but it cannot increase the diversity of the prompts.

3.Evaluation. The safety of generated videos is evaluated by GPT-4 with provided image frames. Is this enough to evaluate the entire video? For example, some harmful frames may not be used for GPT-4 evaluation. Besides, the details of human study are not provided. How many people are involved in the evaluation? Did they get the full video for evaluation or also just some frames?

**Strengths:**

See above.

**Additional Feedback:**

N/A

**Documentation:**

The details of evaluation and some part of the data are provided.

**Ethics:**

1

**Limitations:**

The limitations are provided by the authors.

**Opportunities For Improvement:**

1.A more in-depth analysis and evaluation of unique safety aspects of T2V models such as the Temporal risk would better highlight the paper’s contribution compared to image-based benchmarks.

2.Increasing the data sources or applying frameworks that can enhance the diversity of GPT-4 generated prompts (e.g., Self-Instruction [1]) could provide a broader range of prompts.

3.A better evaluation method is needed to build the gap between GPT-4 based and Human-based assessment (such as the Disturbing Content). The evaluation bias caused by frame-based assessment (compared to evaluating the entire video) also needs to be explained.

[1] Yizhong Wang et al. Self-Instruct: Aligning Language Models with Self-Generated Instructions. ACL 2023.

**Relation To Prior Work:**

Yes.

**Summary And Contributions:**

This paper introduces a benchmark for evaluating the safety of text-to-video (T2V) generative models. The benchmark identifies 12 potential safety aspects in T2V models and uses GPT-4 along with human reviewers to generate prompts for each category. Additionally, it incorporates three types of prompt attacks to further expand the prompt set. For evaluation, the benchmark captures image frame of each second of the generated videos, which are then assessed for safety by GPT-4, and these results are compared with human assessments. The safety of four generative models is evaluated using the proposed benchmark.

---

> ### Author Rebuttal · Authors · 2024-08-26
>
> ***Question 3: Evaluation. Is frame-based assessment by GPT-4 enough to evaluate the entire video? For example, some harmful frames may not be used for GPT-4 evaluation. Besides, the details of human study are not provided.***
>
>
> Thanks for the valuable suggestion.
> We provide additional details of human study as follows.
> We recruit 60 volunteers for the assessment, all of whom must be at least 18 years old, in good physical and mental health, and free from conditions such as heart disease or vasovagal syncope.
> Before the assessment, we present definitions and examples of each type of safety risk to the volunteers.
> Volunteers **view full videos** on 22-24 inch monitors.
> Volunteers are given a 10-minute break after every 20 minutes of review to ensure psychological comfort and sustained attention.
> Each video receives evaluations from at least three volunteers. Following the initial evaluations, we conducted a secondary round of cross-validation.
>
> Additionally, for GPT-4-based evaluation, we sample an image frame per second from each generated video and present these multi-frame images alongside prompts to GPT-4.
> Given that existing models produce videos with only gradual and subtle changes between frames, a one-frame-per-second sampling frequency is sufficient to detect harmful content.
> Actually, the high correlation between GPT-4’s evaluations and human evaluations also validates that the one-frame-per-second sampling frequency for GPT-4 assessment is effective in this context.
> This also represents one of the key findings and contributions of our paper.
>
> The gap between GPT-4 based and human based assessment in the dimension of disturbing content is not due to the evaluation bias caused by frame-based assessment, but rather stems from GPT-4’s limited ability to fully understand scenarios that evoke fear and discomfort in humans without explicit elements like gore.
> A straightforward remedy to better evaluation is using in-context learning by providing examples of disturbing content frames before assessment, which increases the correlation coefficient from 0.589 to 0.727, bridging the gap.
>
> Furthermore, to more convincingly verify the absence of evaluation bias in GPT-4 caused by frame-based assessment, we further conduct new experiments on human study. In these experiments, human evaluators are shown only the sampled frames at a rate of one frame per second, rather than the complete videos.
> The results indicate that there is virtually no difference between the frame-based and full-video evaluations by human raters, which further supports the adequacy of the one-frame-per-second sampling frequency for detecting harmful content.
>
> We will include above discussion in the revision.
>
> [1] VidProM: A Million-scale Real Prompt-Gallery Dataset for Text-to-Video Diffusion Models, arXiv 2024
>
> [2] Self-Instruct: Aligning Language Models with Self-Generated Instructions, ACL 2023

---

> ### Author Rebuttal · Authors · 2024-08-26
>
> **We apologize for the delay in submitting our rebuttal. The delay is due to our efforts to incorporate additional experiments based on the reviewers' feedback, which have only recently been completed.**
> Thank you for acknowledging the novelty of our paper as well as providing the valuable feedback. Below we address the detailed comments, and hope that you can find our response satisfactory.
>
>
>
>
>
>
> ***Question 1: A more in-depth analysis and evaluation of unique safety aspects of T2V models such as the Temporal risk would better highlight the paper’s contribution.***
>
> Thanks for the valuable suggestion.
> We provide a more in-depth analysis of the unique safety risks associated with video.
>
> Existing text-to-image safety benchmarks have limited their focus to insufficient aspects such as pornography, violence, gore, hate, and politics, neglecting other critical safety aspects.
> Our work addresses these gaps by conducting a comprehensive safety assessment across 12 crucial dimensions.
> Importantly, these benchmarks also fail to consider the unique temporal risks associated with video.
> For "Temporal Risk", we propose a general definition. It encompasses any cases where "individual frames might appear harmless but the entire sequence can present unsafe content through continuity between frames".
> "Temporal Risk" can be classified into three types:
> **(1) Sequential Action Risk** (where individual frames extracted from actions are benign, but the continuous sequence of actions is not safe for work),
> **(2) Dynamic Variation Risk** (where unsafe content is revealed through dynamically transforming the target into another object over time),
> **(3) Coherent Contextual Risk** (where unsafe content emerges from the temporal linking of presented elements, such as harmful subtitles).
> The evaluation criteria for each category are consistent: “the entire sequence presents NSFW content through continuity between frames”.
> We will clarify this in the revision.
>
> An interesting observation is that the Pika with a non zero temporal risk profile is also the one that is claimed to be more capable of video generation.
> This phenomenon arises because Pika excels in generating continuous actions and variations unique to video content, such as complex movements, subtitle shifts, and transformations in human forms. In contrast, the other three models display weaker generative capabilities and fail to meet the minimum threshold to produce such risks.
> This observation supports our third conclusion: there is a trade-off between the usability and safety of text-to-video generative models.
> This implies that as models enhance, the risk of generating unsafe content may increase unless explicitly handled.
> We will add a more comprehensive discussion about unique risks of videos in the revision.
>
>
>
>
>
> ***Question 2: Increasing the data sources or applying frameworks that can enhance the diversity of GPT-4 generated prompts could provide a broader range of prompts.***
>
> Thanks for the valuable suggestion.
> We further expand the dataset to enhance prompt diversity by using VidProM [1] and Self-Instruct [2].
> VidProM is a large-scale dataset comprising unique text-to-video prompts from real users.
> We select NSFW prompts from VidProM to supplement T2VSafetyBench, thereby increasing the data sources and better reflecting the prompts in the real-world.
> Additionally, we also employ the Self-Instruct framework with T2VSafetyBench as the seed set (where T2VSafetyBench already includes some prompts from VidProM, thus containing both real-world prompts and LLM-generated prompts), to generate a broader and more diverse range of prompts.
> The updated dataset is available at the following link (https://anonymous.4open.science/r/T2VSafetyBench_Dataset-248D/).
> We will add a more detailed information about dataset construction in the revision.

---

> > ### Comment · Reviewer_53CX · 2024-08-28
> >
> > Thanks for the responses. My concerns about the details and evaluations have been partially addressed. I will raise my score to 6.

---

> > > ### Author Response · Authors · 2024-08-29
> > > **Thank you for increasing the score**
> > >
> > > Dear Reviewer 53CX,
> > >
> > > Thank you very much for increasing the score! We are glad to know that our response has addressed your concerns. We really appreciate your valuable comments and appreciation of our contributions. We will further improve the paper in the final.
> > >
> > > Best regards, Authors

---

### Official Review · Reviewer_4PQa · 2024-07-25
**T2VSafetyBench: Evaluating the Safety of Text-to-Video Generative Models**

**Rating:** 5
**Confidence:** 4
**Clarity:** Yes.

**Review:**

#Strengths

* Trendy Topic.

* This paper considers Temporal Risk, a risk unique to video generation models.

#Weaknesses

* The constructed benchmark may not be comprehensive enough, particularly for a dataset and benchmark paper, as it includes only 4 video generation models. In contrast, existing benchmarks related to text-to-video generative models, such as Sun et al. [1], involve 20 different models. It is recommended to increase the variety of models used to enhance the comprehensiveness of the benchmark and its findings.

* Some statistical information about the proposed benchmark might be missing. For instance, it does not specify how many videos are generated for each dataset, nor does it include the number of frames for each dataset.

* The description of the evaluation process might also lack clarity. For example, the number of human evaluators is not specified. Additionally, for GPT-4 based evaluation, it is unclear whether all frames of the generated video are fed into the GPT-4 model or if only a subset of frames is sampled. If only a subset of frames is used, what specific method is employed to sample these frames, and how can it be ensured that the temporal risk is detected, given that harmful content could potentially be hidden in the unsampled frames?

* It is suggested to include more discussion about the safety mechanisms employed in the involved models. Do these models incorporate safety mechanisms such as safety alignment or safety classifiers for the generated content? Additionally, discussing the influence of these safety mechanisms would be beneficial.

[1] Sun et al., T2V-CompBench: A Comprehensive Benchmark for Compositional Text-to-video Generation. arXiv 2024.

**Strengths:**

See Review.

**Additional Feedback:**

N/A

**Correctness:**

Overall, most of it is correct. Some information about the GPT-4 based evaluation is missing.

**Documentation:**

Some information is missing.

**Ethics:**

No.

**Limitations:**

It is discussed in the Appendix.

**Opportunities For Improvement:**

See Review.

**Relation To Prior Work:**

Yes.

**Summary And Contributions:**

This paper proposes a benchmark designed to assess the safety of images generated by current text-to-video models. The benchmark's text prompts are created by identifying key aspects of video generation safety and developing malicious prompt datasets using LLMs and prompt attacks. The benchmark reveals several insights, such as no single model performing well in all areas, a generally high correlation between GPT-4 assessments and manual reviews, and a trade-off between usability and safety in text-to-video generative models.

---

> ### Author Rebuttal · Authors · 2024-08-26
>
> ***Question 3: The description of the evaluation process might also lack clarity. Additionally, if only a subset of frames is used for GPT-4 based evaluation, how can it be ensured that the temporal risk is detected?***
>
> Thanks for the valuable suggestion.
> We provide additional details on the evaluation process here.
> We recruit 60 volunteers for the assessment, all of whom must be at least 18 years old, in good physical and mental health, and free from conditions such as heart disease or vasovagal syncope.
> Before the assessment, we present definitions and examples of each type of safety risk to the volunteers.
> Each video receives evaluations from at least three volunteers. Following the initial evaluations, we conducted a secondary round of cross-validation.
>
> Additionally, for GPT-4-based evaluation, as detailed in Section 3.3, we sample an image frame per second from each generated video and present these multi-frame images alongside prompts to GPT-4.
> Given that existing models produce videos with only gradual and subtle changes between frames, a one-frame-per-second sampling frequency is sufficient to detect harmful content.
> Actually, the high correlation between GPT-4’s evaluations and human evaluations also validates that the one-frame-per-second sampling frequency for GPT-4 assessment is effective in this context.
> This also represents one of the key findings and contributions of our paper.
>
> Furthermore, to more convincingly verify the absence of evaluation bias in GPT-4 caused by frame-based assessment, we further conduct new experiments on human study. In these experiments, human evaluators are shown only the sampled frames at a rate of one frame per second, rather than the complete videos.
> The results indicate that there is virtually no difference between the frame-based and full-video evaluations by human raters, which further supports the adequacy of the one-frame-per-second sampling frequency for detecting harmful content.
> We will include more detailed information about evaluation process in the revision.
>
>
>
>
>
>
>
> ***Question 4: It is suggested to include more discussion about the safety mechanisms employed in the involved models.***
>
> Thanks for the valuable suggestion.
> We provide an extended discussion about the safety mechanisms employed in the involved models.
>
> Safety filters, also known as safety classifiers, can be categorized into two types: pre-processing safety filter and post-processing safety filter.
> Pre-processing safety filter operates directly on the text itself or its embedding space. Usually, it blocks prompts containing sensitive keywords/phrases in a predefined list or prompts that are close to these sensitive keywords/phrases in the embedding space.
> Post-processing safety filter, on the other hand, operates on the generated videos. Specifically, post-processing safety filters can be binary video classifiers, which predict whether the generated video is sensitive or not.
> Pika employs a pre-processing safety filter; Gen2 uses a post-processing safety filter; SVD incorporates both pre-processing and post-processing safety filters; and Open-Sora does not utilize any safety filter.
> Different safety mechanisms impact model security in various ways.
> Pika and Open-Sora exhibit higher security risks concerning Pornography, Borderline Pornography, Gore, and Disturbing Content. This heightened vulnerability may stem from the lack of post-processing safety filters for videos, resulting in ineffective defenses against these more explicit NSFW dimensions.
> Conversely, Pika demonstrates lower safety risks in the context of Political Sensitivity due to its pre-processing safety filter's ability to identify politically sensitive keywords and subsequently prevent the generation of such videos.
>
> Additionally, safety alignment is also extensively studied in generative models, particularly in large language models.
> However, since safety alignment is applied during the training phase, specific details and processes of safety alignment in the models cannot be determined directly, and its impact can only be inferred from experimental results.
> In terms of Gore, Gen2 achieves optimal performance, with a NSFW rate approaching 0\% and completely no blood present in the generated videos.
> Regarding Copyright and Trademark, Pika exhibits exceptional defensive capability, ensuring that the generated videos are free of infringing marks.
> These outcomes likely stem from safety alignment during the model training process, aligning with the human values of "no blood" or "no infringing marks."
> We will add the discussion in the revision.

---

> > ### Author Response · Authors · 2024-08-31
> > **Looking forward to further feedback**
> >
> > Dear Reviewer 4PQa,
> >
> > Thanks again for appreciating our contributions as well as providing valuable comments. We have carefully addressed them in detail. As the rebuttal is about to close, we hope you may find the response satisfactory (as the other reviewers) and could kindly raise your score, and we are happy to address further feedback (if any).
> >
> > Best regards, Authors

---

> ### Author Rebuttal · Authors · 2024-08-26
>
> **We apologize for the delay in submitting our rebuttal. The delay is due to our efforts to incorporate additional experiments based on the reviewers' feedback, which have only recently been completed.**
> Thank you for appreciating our new contributions as well as providing the valuable feedback. Below we address the detailed comments, and hope that you can find our response satisfactory.
>
>
>
>
>
>
>
> ***Question 1: It is recommended to increase the variety of models used to enhance the comprehensiveness of the benchmark and its findings.***
>
> Thanks for the valuable suggestion and for pointing out the relevant work by Sun et al. [1] that can be referenced.
> We further conduct experiments on 7 additional video generation models, bringing the total to 11.
> The 7 newly tested models are Gen3, Kling, Vidu, Ying (Zhipu), Open-Sora-Plan v1.2.0, OpenSora 1.0, and OpenSora 1.2.
> The results are shown below.
>
> |Aspect|Gen3||Kling||Vidu||Ying (Zhipu)||Open-Sora-Plan v1.2.0||OpenSora 1.0||OpenSora 1.2||
> |:-:|:-:|:-:|:-:|:-:|:-:|:-:|:-:|:-:|:-:|:-:|:-:|:-:|:-:|:-:|
> ||GPT-4|Human|GPT-4|Human|GPT-4|Human|GPT-4|Human|GPT-4|Human|GPT-4|Human|GPT-4|Human|
> Pornography|0.0%|0.0%|2.5%|3.8%|2.8%|4.9%|3.0%|6.9%|27.8%|31.8%|52.5%|45.3%|51.4%|50.7%|
> Borderline Pornography|22.4%|16.2%|1.6%|2.7%|4.2%|5.2%|33.5%|41.8%|13.9%|25.1%|33.9%|26.1%|53.7%|62.7%|
> Violence|62.6%|56.6%|38.3%|45.2%|37.2%|45.0%|48.6%|52.2%|41.7%|50.9%|91.7%|83.7%|82.4%|83.7%|
> Gore|0.0%|2.7%|52.8%|58.7%|17.6%|15.4%|31.6%|28.9%|27.3%|36.4%|63.6%|54.5%|72.4%|73.5%|
> Public Figures|100.0%|100.0%|0.0%|0.0%|48.6%|53.8%|69.3%|71.5%|42.6%|34.7%|90.2%|91.8%|100.0%|91.4%|
> Discrimination|5.0%|21.3%|1.5%|4.6%|11.3%|16.2%|7.8%|16.8%|0.0%|7.1%|21.5%|22.0%|21.5%|28.7%|
> Political Sensitivity|57.7%|62.7%|13.7%|18.6%|18.0%|23.4%|6.3%|12.0%|25.9%|18.8%|37.5%|32.0%|31.4%|25.4%|
> Illegal Activities|60.7%|44.3%|25.7%|27.8%|9.2%|14.3%|47.8%|43.5%|12.8%|12.5%|43.0%|44.9%|38.5%|52.4%|
> Disturbing Content|26.9%|21.7%|71.6%|82.4%|38.8%|52.6%|25.7%|28.0%|25.4%|31.4%|91.7%|75.2%|75.4%|83.1%|
> Misinformation|54.8%|57.1%|16.8%|21.5%|76.2%|82.7%|57.8%|59.1%|37.6%|41.7%|75.6%|65.7%|79.6%|80.5%|
> Copyright and Trademark|59.9%|62.5%|84.2%|89.4%|10.5%|19.0%|85.0%|87.0%|0.0%|0.0%|45.2%|31.8%|42.5%|39.0%|
> Temporal Risk|12.4%|12.9%|49.0%|54.3%|80.1%|74.0%|55.0%|45.5%|0.0%|0.0%|7.3%|6.8%|9.5%|7.5%|
> Average NSFW Rate|38.5%|38.2%|29.8%|34.1%|29.5%|33.9%|39.3%|41.1%|21.2%|24.2%|54.5%|48.3%|54.9%|56.6%|
>
> It can be seen that different models showcase distinct strengths.
> Gen3, Kling, Vidu, and Ying (Zhipu) demonstrate exceptional defensive capabilities against pornography.
> Gen3 maintains the lowest safety risk in the context of gore.
> Ying (Zhipu) shows a lower NSFW rate in political sensitivity.
> Kling achieves nearly flawless performance in managing content related to public figures, reaching an almost 0% NSFW rate.
> Vidu performs exceptionally well in handling illegal activities and copyright and trademark issues.
> The safety of Open-Sora-Plan v1.2.0 paradoxically increases due to its relatively weaker generative capabilities and limited understanding (as discussed in our third conclusion).
> Additionally, the consistency between GPT-4 evaluations and human assessments remains high, aligning with our second conclusion.
> Vidu’s superior generative capability in representing continuous actions and changes results in higher temporal risk, supporting our third conclusion.
> These new experimental results enhance the comprehensiveness of the benchmark and its findings.
> We will include more comprehensive experimental results in the revision.
>
> [1] T2V-CompBench: A Comprehensive Benchmark for Compositional Text-to-video Generation, arXiv 2024.
>
>
>
>
>
>
>
>
> ***Question 2: Some statistical information about the proposed benchmark might be missing.***
>
> Thanks for the valuable suggestion.
> For each prompt in our dataset, we generate four videos given a T2V model. Each video generated by Open-sora contains 128 frames, while those produced by other models contain 96 frames each.
> The resolutions of videos generated by Pika, Gen2, SVD, and Open-sora are 1280 × 720, 1152 × 640, 1024 × 576, and 848 × 480, respectively.
> We will include more detailed information in the revision.

---

### Official Review · Reviewer_JP3C · 2024-08-03
**Paper describing a safety benchmark and corresponding evaluation methodology for T2V models**

**Rating:** 7
**Confidence:** 3

**Review:**

The paper addresses the problem of safety in T2V models.

Pros:
As the generative capabilities of such models improve the safety of their output will become more and more relevant and benchmarks will be very important to evaluate them. The work is therefore very significant for this track.

The paper is well structured, clear in the exposition and it should overall allow any reader to reproduce the evaluation methodology from scratch.

Several openly available models are studied with the benchmark, which provide a good perspective on the safety of T2V systems (according to the proposed aspects)

The paper conducts an in depth comparison between human and GPT-4 as safety evaluators of the generated media as a validation of their evaluation system based on GPT-4

Cons:

Of the 12 aspects of safety only the last one considers the video aspect of the generated output: the "Temporal risk". This seems to be a very generic aspect, compare to the other 11 ones which are more clearly defined. I wonder if it would be possible to better describe individual action performed by people or their interaction in the video.It's also interesting that the only T2V model with a non zero temporal risk profile is also the one that is claimed to be more capable of video generation.

There is a connection between generation ability of a model and its safety, there is no indication if the generated videos are of good quality in term of prompt fidelity and overall appearance. A study across the safety vs generation quality (both for individual frames and temporal consistency) is probably necessary. In general, I think the benchmark should always be used with a corresponding quality measurement.

**Strengths:**

The paper addresses an important problem in the field of generative media: the safety of T2V. It provides a useful framework for evaluating the safety of T2V.  It is clearly written and explain all the relevant steps in creating the benchmark. It also provides a validation of the safety evaluation system based on GPT-4

**Additional Feedback:**

I think a deeper discussion of how the generation quality of a model impacts its safety and how the two should be study together will be very useful. A more fine grain categorization of the temporal risk could also be useful.

**Clarity:**

Yes, the paper is clearly written and provide all the relevant information for reproducing the benchmark.

**Correctness:**

The paper is appears correct. It is mainly descriptive of an evaluation methodology.

**Documentation:**

I could not find a list of the generated prompts in the supplemental material. The authors provide the GPT-4 prompt used to generate them. There are no jail-breaking prompts.

**Ethics:**

The paper addresses issues of safety and should undergo an Ethics in particular for the specific safety aspects proposed. From my perspective there are no other ethical issues. Only the specific topic.

**Limitations:**

Yes the authors have addessed the limitation of their work and possible negative impact in their supplemental material.

**Opportunities For Improvement:**

Coupling both safety and a measure for quality would provide a more insightful analysis for this benchmark. Possibly just consider an already available quality metric.

**Relation To Prior Work:**

The paper describes the connection between the T2V safety problem and proposed benchmark and other measurement of quality for T2V models. It also describes connections between the safety approach adopted and other safety benchmark introduced for other generative models.

**Summary And Contributions:**

This paper considers the issue of safety for T2V models. It defines 12 ***aspects*** relevant for the safety of a T2V model:
 * Pornography,
 * Borderline Pornography (could be redefined as sexualization),
 * Violence,
 * Gore,
 * Public Figures,
 * Discrimination,
 * Political Sensitivity,
 * Illegal Activities
 * Disturbing Content
 * Misinformation and Falsehoods,
 * Copyright and Trademark Infringement
 * Temporal Risk

Of the 12 aspects only the last one is specific of video and it seems to capture how a video could use the temporal change to violate one of the previous aspects, even though each individual frame would still be considered safe.

To evaluate a model along the 12 aspects, the author used both LLM models and prompt jail-breaking methods to create a set of malicious prompts. For each prompt and generated video (sampled at 1 frame per second) they construct a new query for GPT-4 to assess for the corresponding safety aspects (see Fig3 in the paper). GPT4 returns a Yes/No answer with a corresponding probability. Finally to validate the precision of GPT-4 as an evaluator of risk the authors also have a sample of the queries (prompt, video) evaluated by human taggers and report the correlation between GPT-4 and the humans.

Several t2v models are studied in the paper as a concrete example of how to use the benchmark as well as providing a perspective on the current status of T2V safety. The model considered are: Pika [1], Gen2 [9], SVD [5], and Open-Sora [16].

The key findings of the paper are:

 * For the 12 aspects of safety no model has good results across the board.
 * The human vs GPT-4 correlation is quite high for all aspects, exceeding 80% in most of them. This suggest that GPT-4 is a reliable evaluator.
 * Less capable is a model from a generative perspective the safer it appears. This implies that as models improve their safety risks will increase unless explicitly handled.

---

> ### Author Rebuttal · Authors · 2024-08-26
>
> **We apologize for the delay in submitting our rebuttal. The delay is due to our efforts to incorporate additional experiments based on the reviewers' feedback, which have only recently been completed.**
> Thank you for acknowledging the significance of our paper as well as providing the valuable feedback. Below we address the detailed comments, and hope that you can find our response satisfactory.
>
>
>
>
>
>
> ***Question 1: "Temporal risk" seems to be a very generic aspect, compare to the other 11 ones which are more clearly defined. I wonder if it would be possible to better describe individual action performed by people or their interaction in the video. It's also interesting that the only T2V model with a non zero temporal risk profile is also the one that is claimed to be more capable of video generation.***
>
> Thanks for the valuable suggestion.
> Indeed, our definition of "Temporal Risk" is currently general. It encompasses any cases where "individual frames might appear harmless but the entire sequence can present unsafe content through continuity between frames".
> Several types of temporal risks can be identified:
> **(1) Sequential Action Risk** (where individual frames extracted from actions are benign, but the continuous sequence of actions is not safe for work),
> **(2) Dynamic Variation Risk** (where unsafe content is revealed through dynamically transforming the target into another object over time),
> **(3) Coherent Contextual Risk** (where unsafe content emerges from the temporal linking of presented elements, such as harmful subtitles).
> We will clarify and give a more fine grain categorization in the revision.
>
> An interesting observation is that the Pika with a non zero temporal risk profile is also the one that is claimed to be more capable of video generation.
> This phenomenon arises because Pika excels in generating continuous actions and variations unique to video content, such as complex movements, subtitle shifts, and transformations in human forms. In contrast, the other three models display weaker generative capabilities and fail to meet the minimum threshold to produce such risks.
> This observation supports our third conclusion: there is a trade-off between the usability and safety of text-to-video generative models.
> This implies that as models enhance, the risk of generating unsafe content may increase unless explicitly handled.
>
>
>
>
>
>
> ***Question 2: Coupling both safety and a measure for generation quality would provide a more insightful analysis for this benchmark.***
>
> Thanks for the insightful suggestion.
> We further evaluate the generation quality of text-to-video models Pika, Gen2, and SVD from the perspectives of text relevance and temporal coherence.
> For text relevance, we use UMTScore [1] to measure the correlation between prompts and videos, where a higher score indicates better performance.
> For temporal coherence, we use CHScore [2] to assess the smoothness and logical sequence of video content over time, with higher scores being preferable.
> In terms of text relevance, the UMTScore of Pika, Gen2, and SVD are 2.443, 2.176, and 2.217, respectively, with Pika exhibiting the highest prompt fidelity.
> This result supports our third conclusion: there is a trade-off between the usability and safety of text-to-video generative models.
> Stronger generative capability (Pika's highest UMTScore) paradoxically correlates with higher safety risks (Pika's highest Average NSFW Rate).
> In terms of temporal coherence, the CHScore of Pika, Gen2, and SVD are 4.00, 5.27, and 4.39, respectively, showing similar performance across the models  (CHScore differences of about 1 are not significant, see [2]).
> Intuitively, the causal relationship between temporal coherence and safety is minimal, as the smoothness of video content over time does not affect the ability to generate NSFW content but influences the perceived fluidity of the video.
> We will include more comprehensive experimental results and discussions on safety vs generation quality in the revision.
>
>
>
>
>
>
> ***Question 3: I could not find a list of the generated prompts in the supplemental material. The authors provide the GPT-4 prompt used to generate them. There are no jail-breaking prompts.***
>
> We have provided the generated prompts in the data-1063 folder within the supplemental material, rather than in the supp-1063.pdf.
>
> ***Question 4: The paper addresses issues of safety and should undergo an Ethics in particular for the specific safety aspects proposed. From my perspective there are no other ethical issues. Only the specific topic.***
>
> We have provided a comprehensive Ethical Considerations in response to the comments from the two Ethics Reviewers.
>
> [1] FETV: A Benchmark for Fine-Grained Evaluation of Open-Domain Text-to-Video Generation, NeurIPS 2023 Datasets and Benchmarks Track
>
> [2] ChronoMagic-Bench: A Benchmark for Metamorphic Evaluation of Text-to-Time-lapse Video Generation, arXiv 2024

---

### Author Response · Authors · 2024-09-01
**Summary of Rebuttal**

Dear Reviewers, AC, and SAC:

We deeply thank the work done by AC and SAC such as distributing the paper to reviewers, guiding the reviewing process, and further supervising the discussion. We also sincerely appreciate the reviewers for taking the time to read our paper, providing constructive comments, and getting involved in our discussion. Without your elaborative help and support, our paper could not have been further polished.

Here we summarize our rebuttal to present a general perspective which could hopefully help grasp our contribution quickly.

Through interactive discussion, four reviewers have **found our response satisfactory/agreed to accept our paper**, and one reviewer has not provided further feedback on our rebuttal.

The final scores given by the reviewers are as follows:
- **Rating: 7, 5, 6, 6, 7** (Reviewer zpBN is happy to raise the score to 7, but since the editing option is closed on Aug 30, they have written the raising score to 7 in the Official Comment)

Additionally, several consensuses have been achieved:
- **The presentation of our paper is clear, well structured, and easy to understand.** (Reviewers JP3C, 53CX, sQtg, zpBN)
- **Our work and addressed problem are trendy, important, and have significant impact** for the track of generative model safety. (Reviewers JP3C, 4PQa, 53CX)
- **Our research perspective and findings are novel, interesting, and make sense.** (Reviewers JP3C, 53CX, sQtg, zpBN)
- **The experimental evaluation and discussion are depth, extensive and comprehensive,** with both GPT-4 and human studies provided. (Reviewers JP3C, 53CX, sQtg)

In the past few weeks, we have tried our best to improve the quality of this paper and address each concern from all reviewers. We sincerely hope our effort can contribute to the community. Thanks again for your kind help and constructive opinions, we are truly grateful to have advice from you.

Sincerely,
Authors.

---

### Decision · Program_Chairs · 2024-09-26

**Decision:**

Accept (Poster)

**Comment:**

**Paper Summary:**
This paper introduces T2VSafetyBench, a benchmark for evaluating the safety of text-to-video (T2V) generative models. The authors define 12 critical aspects of video generation safety, construct a malicious prompt dataset, and evaluate a number of T2V models. Key findings include: no single model excels in all aspects, GPT-4 assessments correlate well with manual reviews, and there's a trade-off between usability and safety in T2V models.

**Pros:**
* First known benchmark with  focus on safety evaluation for T2V models
* Comprehensive benchmark covering 12 safety aspects, including unique temporal risks
* Interesting findings on model performance and automated assessment methods
* Potential impact on understanding and improving T2V model safety

**Cons:**
* Initial concerns about dataset size and diversity (addressed in rebuttal)
* Some reviewers noted limited novelty compared to text-to-image safety evaluations
* Ethical concerns regarding exposure to disturbing content and potential biases (discussed in Ethics Review section)

**Review Summary:**
Reviewers generally appreciated the importance and timeliness of the work. Some reviewer’s concerns about dataset size and diversity were addressed during the rebuttal period, with the authors expanding the dataset and incorporating real-world prompts. The correlation between GPT-4 and human assessments to validate the use of GPT-4 as evaluator was seen as a valuable contribution. Some reviewers questioned the novelty compared to text-to-image safety evaluations, but the authors clarified the unique aspects of video safety, in particular temporal risks. Ethical concerns were raised and addressed through the authors' detailed explanation of safeguards and sharing of IRB approval (see below).

**Ethics Review:**
Two ethics reviewers raised concerns about human exposure to disturbing content and potential biases. The authors provided a comprehensive response, detailing safeguards for human evaluators, IRB approval, and measures to mitigate bias, which addressed the concerns of the ethics reviewers.

**AC Recommendation:** Accept

**Rationale:**
The paper is well structured and easy to follow. The focus on T2V safety is timely and important, with the benchmark providing a valuable tool for the research community. Despite some initial concerns, the authors have adequately addressed most issues in their rebuttal. The expanded dataset, incorporation of real-world prompts, and increased number of benchmarked models strengthen the paper's contribution. The ethical concerns were addressed through appropriate safeguards and IRB approval. The paper's findings, particularly the trade-off between model capability and safety, offer insights for future T2V model development. While some limitations remain, the overall contributions and potential impact of this work justify its acceptance.